# The Combination of Hearing Impairment and Frailty Is Associated with Cognitive Decline among Community-Dwelling Elderly in Japan

**DOI:** 10.3390/ijerph20054437

**Published:** 2023-03-02

**Authors:** Akie Kawamura, Naoto Kamide, Masataka Ando, Takeshi Murakami, Machiko T. Shahzad, Kayoko Takahashi

**Affiliations:** 1School of Allied Health Sciences, Kitasato University, 1-15-1 Kitazato, Minami-ku, Sagamihara 252-0373, Japan; 2Graduate School of Medical Sciences, Kitasato University, 1-15-1 Kitazato, Minami-ku, Sagamihara 252-0373, Japan; 3School of Nursing, Kitasato University, 1-15-1 Kitazato, Minami-ku, Sagamihara 252-0373, Japan

**Keywords:** cognitive decline, community-dwelling older adult, frailty, hearing impairment, pre-frailty

## Abstract

Hearing impairment and frailty are associated with cognitive decline in older people. This study aimed to investigate the effect of the interaction between hearing impairment and frailty on cognitive decline in community-dwelling older people. A mail survey of community-dwelling, older people (age ≥ 65 years) who lived independently was conducted. Cognitive decline was defined using the self-administered dementia checklist (≥18 out of 40 points). Hearing impairment was assessed using a validated self-rated questionnaire. Furthermore, frailty was assessed using the Kihon checklist, and robust, pre-frailty, and frailty groups were identified. Multivariate logistic regression analysis, adjusted for potential confounding factors, was performed to determine the association of the interaction between hearing impairment and frailty with cognitive decline. Data obtained from 464 participants were analyzed. Hearing impairment was independently associated with cognitive decline. Additionally, the interaction term of hearing impairment and frailty was significantly related to cognitive decline. For participants in the robust group, hearing impairment was not associated with cognitive decline. In contrast, for participants in the pre-frailty or frailty groups, hearing impairment was associated with cognitive decline. The association between hearing impairment and cognitive decline was affected by frailty status in community-dwelling, older people.

## 1. Introduction

Cognitive decline is a major age-related global health problem. When cognitive decline leads to disability in activities of daily living, the diagnosis of dementia is given. Globally, an estimated 57.4 million people were living with dementia in 2019, and this figure is expected to rise sharply to 152.8 million by 2050 [1]. Japan has become the world’s leading super-aged society, and the estimated prevalence of dementia in 2012 among people aged 65 years and older was 15%, and it exceeded 30% among those aged 80 years and older. This prevalence is expected to continue increasing in the future [2]. Thus, the establishment of preventive measures against cognitive decline is an important issue for maintaining and improving the quality of life (QOL) of older adults.

There are many known potentially modifiable risk factors for cognitive decline leading to dementia [3,4]. It is, therefore, crucial to identify modifiable risk factors for cognitive decline and to develop effective preventive measures. Hearing impairment has attracted attention in recent years as a potentially modifiable risk factor for cognitive decline. Age-related hearing impairment in older persons, in particular, is known to greatly affect communication, activities of daily living, and social activities of older adults [5,6]. In fact, many studies have shown evidence of the association between hearing impairment and cognitive decline. A cohort study of community-dwelling older persons in the United States found that hearing impairment was associated with cognitive decline, a precursor to Alzheimer’s disease [7,8]. Similarly, a cohort study in the United Kingdom targeting residents aged 50 years and older reported an association between hearing impairment and cognitive decline. It especially highlighted the fact that the coexistence of hearing impairment and social isolation has the most significant impact on cognitive decline [9]. A report from the Lancet International Commission, which provides a model of factors that could prevent the onset of dementia through medical interventions, stated that, ultimately, hearing impairment beyond middle age is the leading factor for which medical intervention can prevent the onset of dementia [3,4].

Besides hearing impairment, frailty is also believed to be an important potentially modifiable risk factor for cognitive decline. Frailty is a condition where, due to an age-related decline in reserve strength, capacity to recuperate from stress is diminished, making individuals more prone to outcomes such as needing nursing care or even causing death, and it is positioned as the preliminary step before needing nursing care [10]. A previous cross-sectional study of community-dwelling older persons suggested that frailty was strongly associated with cognitive decline and dementia [11]. Another cohort study of community-dwelling older adults in the United States found that frailty was not only related to the risk of developing mild cognitive impairment (MCI), which is the pre-dementia stage, but that frailty also accelerated cognitive decline [12]. A meta-analysis also showed that frailty was a major predictive factor of Alzheimer’s disease, vascular dementia, and all dementias, among community-dwelling older people [13].

As mentioned above, hearing impairment and frailty have been shown to be potentially modifiable risk factors for cognitive decline in community-dwelling older adults. Meanwhile, many studies have reported the association between hearing impairment and frailty. Other studies on community-dwelling older persons have also reported that hearing impairment was a risk for exacerbating frailty [14,15,16,17]. Thus, an interaction may exist between hearing impairment and frailty, and, as such, the influences of both need to be considered when verifying their associations with cognitive decline. However, no studies have examined the association of both hearing impairment and frailty with cognitive decline in independent community-dwelling older adults. Early preventive intervention for cognitive decline warrants the examination and identification of the association of both hearing impairment and frailty with cognitive decline. Thus, the present research was conducted as a cross-sectional study targeting independent community-dwelling older persons to investigate the association of hearing impairment and frailty with cognitive decline.

## 2. Materials and Methods

### 2.1. Participants

A cross-sectional study using a mail-in, self-administered questionnaire survey was conducted between October and November 2021, targeting older persons aged 65 years and older who lived in an apartment complex in Sagamihara City, Kanagawa Prefecture, Japan. A total of 2117 older persons were living in the apartment complex as of 1 January 2021, and surveys were mailed to all older adults. The inclusion criteria for the present study consisted of the following: (i) being able to live independently; and (ii) living in the community. The exclusion criteria consisted of the following: (i) being assigned a care level or support level by the Japanese long-term care insurance; and (ii) living in a nursing home. In the long-term care insurance system in Japan, persons with a defined dementia must be certified as care need level 1 or higher. The participants in this study excluded those who are certificated of long-term care need, and therefore exclude those with a confirmed diagnosis of dementia.

### 2.2. Measurements

#### 2.2.1. Cognitive Decline

The self-administered dementia checklist (SDC) [18,19] was used. This checklist is a self-administered questionnaire that uses a 4-point Likert scale and consists of 10 items related to forgetfulness and insights into activities of daily living. It is a scale with confirmed validity that detects cognitive decline in the early stages of dementia in older adults. The assessment score ranges between 10 and 40 points, with a higher number indicating a higher likelihood of cognitive decline. Based on previous studies [18,19], the present study defined cognitive decline as a score of 18 points or higher.

#### 2.2.2. Frailty Status

Frailty status was determined using the Kihon checklist (KCL) [20]. The KCL is a comprehensive assessment tool that consists of 25 questions in the following seven domains: (i) 5 questions to assess movements related to activities of daily living; (ii) 5 questions to assess locomotive function; (iii) 2 questions to assess poor nutritional status; (iv) 3 questions to assess oral function; (v) 2 questions to assess social withdrawal; (vi) 3 questions to assess cognitive function; and (vii) 5 questions to assess depressive mood. Each question is given a score of 0 or 1, and the score of all questions is totaled (between 0 and 25). According to a systematic review, the KCL is a reliable tool for assessing frailty in older persons [21]. The following definitions were used in the present study: frailty (8 points and above); pre-frailty (between 4 and 7 points); and robust (between 0 and 3 points) [22,23].

#### 2.2.3. Hearing Impairment

Six questions under the hearing impairment subcategory of Suzuki et al.’s The Questionnaire for Hearing [24,25] were used as the self-assessment scale for assessing hearing impairment related to activities of daily living. The relationship between this scale and objective auditory function has been confirmed [24,25], validating its use to assess hearing impairment. Each question was scored from 1: “Always audible” to 5: “Always inaudible”, depending on the degree of hearing impairment, and the mean score was calculated. The operational definition used in this study was that a score of 1 was defined as no hearing impairment. The higher the score, the more likely there is a hearing impairment.

#### 2.2.4. Confounding Factors

The following were investigated: age, sex, height, weight, body mass index, financial situation, presence or absence of cohabiting family members, social isolation, social participation, subjective health, medical history, medication status, and nutritional intake status.

To assess their financial situation, the participants were asked about financial security to enjoy their hobbies and little luxuries. The responses were scored on a 4-point Likert scale: “plenty”, “some”, “not much”, and “none”. For the analysis, the results were categorized into the following two binary responses: “plenty” and “some” were categorized under having financial security, and “not much” and “none” indicated financial insecurity. To assess social isolation, the Japanese version of the abbreviated 6-item Lubben Social Network Scale [26] was used. The scale consists of six questions related to family networks and friendship (not family) networks, and it is assessed with a score from 0 to 30 points. A lower score indicates poorer social networks, and 11 points or below was used to determine social isolation. Social participation was classified into six types of activities involving participation: (i) community events, (ii) neighborhood and residents’ associations, (iii) activities in senior groups, (iv) activities related to one’s hobbies, (v) volunteer activities, and (vi) activities related to handing down tradition. Participation status in these activities was assessed with a score from 0 to 6 [27]. Subjective health was evaluated with a 6-point Likert scale from 1 (Excellent) to 6 (Very poor) points, with 1 to 3 points indicating “being healthy” and 4 to 6 points showing “poor health”. These were used in subsequent analyses. Past medical history consisted of determining the presence or absence of hypertension, diabetes mellitus, and cardiovascular disease. As for medication status, polypharmacy was defined as taking five medications or more. Nutritional intake status was evaluated using the Dietary Variety Score [28]; it was scored from 0 to 10, with a higher score indicating better nutritional intake status. The present study defined good nutritional intake status as having a score of 4 points or higher [29,30,31]. These scores were used in the analysis.

### 2.3. Statistical Analysis

For the surveyed items in the present study, means and standard deviations were calculated for continuous variables, and frequencies were calculated for categorical variables. Subsequently, for each of the surveyed items, differences related to the presence or absence of cognitive decline were analyzed using the unpaired t-test for continuous variables and the chi-squared test for categorical variables. To analyze the associations of cognitive decline with frailty and hearing impairment, a multivariate binomial logistic regression analysis was performed using cognitive decline as the dependent variable and frailty and hearing impairment as the independent variables. Potential confounding factors such as base attributes, socioeconomic factors, and health status were used as adjustment variables. In addition, taking into account the interaction between frailty and hearing impairment, a multivariate binomial logistic regression analysis was conducted using cognitive decline as the dependent variable, and the interaction term between frailty and hearing impairment as the independent variable. All potential confounding factors were used as adjustment variables. The goodness-of-fit of the binomial logistic regression analysis was evaluated using the Hosmer–Lemeshow test and c statistics. Statistical analysis was conducted using R Statistical Analysis Software Version 4.0.3 (R Foundation for Statistical Computing, Vienna, Austria) [32], with significance at 5%.

### 2.4. Ethical Considerations

Information, such as the purpose of the survey and its voluntary nature, was clearly stated in the letter of request and description documents that were enclosed with the survey to the participants. Consent was implied with a returned completed survey. The present study was approved by the Institutional Review Board of the School of Allied Health Sciences at Kitasato University (approval number 2021-026).

## 3. Results

### 3.1. Characteristics of Participants

Responses were obtained from 559 individuals (response rate 26.4%). After excluding those who were certified to receive long-term care and those who had missing data, responses from 464 individuals were used for the analyses. Table 1 is a summary of the results of the analyzed participants. Cognitive decline was found in 18 participants (3.9%), frailty in 100 participants (21.6%), and pre-frailty in 167 participants (36.0%). Although the mean score for hearing difficulty was 1.8 ± 3.8 points, 83 older persons (17.9%) scored 1 point (no problem at all in all the questions).

The results obtained by examining differences related to the presence or absence of cognitive decline showed significant differences in the following: age (*p* = 0.026), social isolation (*p* = 0.005), social participation (*p* = 0.004), subjective health (*p* = 0.01), hearing difficulty (*p* ≤ 0.001), KCL (*p* ≤ 0.001), and frailty (*p* ≤ 0.001). In other words, older persons with cognitive decline were more likely to have (i) a higher age; (ii) a higher percentage of social isolation; (iii) low social participation; (iv) poor subjective health; (v) a higher score for hearing difficulty; and (vi) more frailty.

### 3.2. Relationships of Cognitive Decline with Hearing Difficulty and Frailty

Table 2 and Table 3 show the results from analyzing the associations of hearing impairment and frailty with cognitive decline. When hearing impairment and frailty were set separately as independent variables to analyze their associations with cognitive decline, the results showed that both hearing impairment and frailty were significantly related to cognitive decline, even after adjusting for the effects of potential confounding factors (Table 2). In other words, the higher the score for hearing impairment (odds ratio [OR] = 4.09, 95% confidence interval [C.I.]: 1.85–9.03) and the greater the degree of being frail compared to the robust group (OR = 20.04, 95% C.I. 1.48–280.00), the higher the odds for developing cognitive decline. There was no significant association between pre-frailty and cognitive decline. Moreover, when the association of the interaction between the hearing impairment score and frailty status (frailty, pre-frailty, robust) with cognitive decline was analyzed, the results showed that the interaction term of hearing impairment and frailty was significantly related to cognitive decline, even after adjusting for the effects of potential confounding factors (Table 3). In addition, there was no significant association between hearing impairment and cognitive decline in the robust group. On the other hand, in the pre-frailty and frailty groups, hearing impairment was significantly associated with cognitive decline. In other words, in the pre-frailty and frailty groups, the higher the score for hearing impairment, the higher the odds were for developing cognitive decline (pre-frailty; OR = 2.74, 95% C.I. 1.24–6.03/frailty; OR = 6.20, 95% C.I. 2.54–15.10). Even when frailty was treated as a continuous variable (the score on the KCL) rather than a categorical variable (frailty/pre-frailty/robust), the results did not change. That is, both hearing impairment and KCL score/the interaction term of hearing impairment and KCL score were significantly associated with cognitive decline.

## 4. Discussion

The present study conducted a cross-sectional examination of the associations of hearing impairment and frailty status with cognitive decline in independent, community-dwelling, older adults. Since the present study was not a face-to-face study, standardized cognitive function tests such as the Mini-Mental State Examination (MMSE) could not be administered. Thus, the SDC, a subjective assessment of cognitive decline, was used because it has been shown to be correlated with the results of valid neuropsychological tests such as the MMSE and the Frontal Assessment Battery [18,19], indicating that it is a valid assessment of cognitive decline. A cohort study of older adults showed that subjective memory decline was related to cognitive decline nearly two decades later among older women [33]. A systematic review of prospective longitudinal studies also showed that subjective assessments of cognitive decline were associated with the onset of MCI and dementia [34]. These study findings provide evidence that even subjective assessments of cognitive decline reflect actual cognitive decline. Likewise, subjective assessments are used for hearing impairment. The scale used in the present study has been shown to be correlated with objective hearing levels [24,25], indicating that the score of the scale reflects the actual conditions of decreased hearing. Thus, it is unlikely that the results of the present study will be different from those obtained from cognitive decline and hearing assessments by objective tests, such as neuropsychological tests and hearing tests. Additionally, the subjective assessment of hearing impairment has the advantage of providing a simple way to assess hearing impairment in their daily living.

The present study showed that hearing impairment and frailty were each significantly associated with cognitive decline. Previous studies of community-dwelling older adults have shown separately the association between hearing impairment and cognitive decline [7,8], and the association between frailty and cognitive decline [11,12]. However, the association between cognitive decline and both hearing impairment and frailty had not been identified. The present study showed that both hearing impairment and frailty are factors that are independently related to cognitive decline. In addition, it was found that there is an interaction between hearing impairment and frailty, and that hearing impairment is not associated with cognitive decline in robust older persons. Therefore, we believe that the association between hearing impairment and cognitive decline is influenced by frailty status. Especially in community-dwelling older persons, the coexistence of hearing impairment and frailty may be risk factors for cognitive decline. A study of older adults with MCI reported that having frailty and hearing impairment together was more likely to reduce cognitive function than hearing impairment alone [35]. Although the present study differs from previous studies in that it used a subjective assessment, the results of the present study hold great significance in that they showed that the coexistence of hearing impairment and frailty in independent, community-dwelling older adults could become a risk factor for cognitive decline.

According to a cohort study that investigated the association between hearing impairment and cognitive decline of individuals 50 years and older living in the United Kingdom, social isolation had the greatest impact on cognitive decline in hard-of-hearing persons [9]. A previous study also reported that social isolation is associated with cognitive decline in older persons [36]. The present study also investigated social isolation using a social network scale. However, although univariate analysis showed an association between cognitive decline and social isolation, social isolation was not related to cognitive decline on multivariate analysis. It was only associated with hearing impairment and frailty. Social isolation has been shown to be associated with frailty [37], and, as such, perhaps in the present study, the influence of frailty may have been greater than that of social isolation. At least, the influence of hearing impairment and frailty should be considered when verifying the association between social isolation and cognitive decline.

As a clinical implication for cognitive decline prevention interventions for community-dwelling older adults, the present study identified that the assessment and intervention of hearing impairment and frailty also need to be conducted at the same time. Especially in frail older persons with hearing impairment, interventions such as using hearing aids on a needs basis may be effective. In fact, a cohort study of community-dwelling older persons in France reported that the use of hearing aids attenuated cognitive decline [38]. Another study found significant improvement in working memory when hard-of-hearing subjects used hearing aids [39]. Prevention and improvement of frailty may also be effective when there is hearing impairment. In fact, frailty is a modifiable factor, and appropriate interventions could improve frailty [40]. In the prevention of cognitive decline, conducting assessments and interventions focused on cognitive decline along with assessments and interventions for hearing impairment and frailty may be effective preventive measures.

The present study had a few limitations. First, since this was a cross-sectional study, no clear conclusion can be made about the causal relationships among the three factors: hearing impairment, frailty, and cognitive decline. Second, the present study used subjective indicators to assess cognitive decline and hearing impairment. Although the validity of the scales has been confirmed, the possibility that the results may differ from those obtained using objective assessments cannot be completely excluded. Third, though the present study focused on the interaction between hearing impairment and frailty, visual impairment has also been reported as a risk factor for frailty [41]. Thus, other factors such as attention deficit also need to be considered in future research. Fourth, the possibility that there may have been characteristic biases of the study participants cannot be ruled out. The participants of the present study were older adults leading independent lives in the community, and 3.9% had cognitive decline. The prevalence rates of dementia and MCI in Japan are estimated to be 15% and 13%, respectively [2]. It is possible that the participants of the present study may have been a biased group in good health, influenced by the survey method which involved movement and motivation, as surveys were placed in the mailboxes of participants.

## 5. Conclusions

A cross-sectional study examined the associations of hearing impairment and frailty status with cognitive decline in community-dwelling older adults. The results showed that hearing impairment and frailty were each independently related to cognitive decline. However, the association between hearing impairment and cognitive decline was observed only in frail and pre-frail older persons, and not among robust older persons. Assessments of both hearing impairment and frailty are important for preventing cognitive decline and dementia. In the case of frail and pre-frail older adults, auditory assessments and interventions may need to be considered in addition to improving frailty status.

## Figures and Tables

**Table 1 ijerph-20-04437-t001:** Characteristics of participants and the results of comparisons by cognitive decline.

		All Sample	Cognition		*p* Value *
		n = 464	Normal (n = 446)	Decline (n = 18)	
Age (y)	Mean (SD)	76.3 (6.0)	76.1 (6.0)	79.4 (7.6)	0.026
≥75 y	Number (%)	258 (55.6)	246 (55.2)	12 (66.7)	0.471
Body mass index (kg/m^2^)	Mean (SD)	22.4 (3.1)	22.4 (3.1)	23.1 (3.0)	0.309
Sex (female)	Number (%)	280 (60.3)	273 (61.8)	7 (38.9)	0.089
Financial situation (poor)	Number (%)	106 (22.8)	99 (22.3)	7 (38.9)	0.177
Living arrangement (alone)	Number (%)	144 (31.0)	142 (31.8)	2 (11.1)	0.109
Social isolation (<12 points in 6LSNS)	Number (%)	171 (36.9)	159 (37.2)	12 (75.0)	0.005
Number of social activities (0–6 type)	Mean (SD)	1.9 (1.7)	1.9 (1.7)	0.7 (1.6)	0.004
Subjective health (poor)	Number (%)	122 (26.3)	112 (25.3)	10 (55.6)	0.01
Medical history: Hypertension	Number (%)	213 (45.9)	205 (46.0)	8 (44.4)	>0.99
Medical history: Diabetes mellitus	Number (%)	70 (15.1)	65 (14.6)	5 (27.8)	0.231
Medical history: Cerebrovascular disease	Number (%)	5 (1.1)	4 (0.9)	1 (5.6)	0.476
Polypharmacy (≥5 types of prescribed medicines)	Number (%)	88 (19.0)	81 (19.2)	7 (38.9)	0.082
Nutritional intake status (good)	Number (%)	265 (57.1)	256 (58.6)	9 (50.0)	0.631
Hearing impairment (1–5 points)	Mean (SD)	1.8 (0.8)	1.8 (0.8)	3.0 (0.8)	<0.001
The score of KCL (0–25 points)	Mean (SD)	5.0 (3.8)	4.7 (3.5)	11.8 (5.5)	<0.001
Pre-frailty	Number (%)	167 (36.0)	164 (38.0)	3 (16.7)	<0.001
Frail	Number (%)	100 (21.6)	86 (19.9)	14 (77.8)	
SDC score (10–40 points)	Mean (SD)	12.7 (2.7)			
Cognitive decline (≥18 points in SDC)	Number (%)	18 (3.9)			

* Comparison between the normal group and the cognitive decline group by the *t*-test or the chi-squared test. KCL: Kihon checklist, SDC: Self-administered dementia checklist.

**Table 2 ijerph-20-04437-t002:** Associations of hearing impairment and frailty status with cognitive decline.

Variables	OR	95% C.I.	*p* Value
Age (y)	1.03	0.92–1.15	0.6370
Body mass index (kg/m^2^)	1.23	0.97–1.56	0.0862
Sex (female)	0.65	0.13–3.16	0.5930
Financial status (poor)	0.74	0.17–3.26	0.6900
Living arrangement (alone)	0.27	0.04–2.00	0.1990
Social isolation (<12 points in 6LSNS)	1.55	0.29–8.39	0.6080
Number of social activities (0–6 types)	0.66	0.37–1.18	0.1650
Subjective health (poor)	0.54	0.11–2.68	0.4490
Medical history: Hypertension	1.04	0.23–4.65	0.9600
Medical history: Diabetes mellitus	1.58	0.30–8.22	0.5860
Medical history: Cerebrovascular disease	3.42	0.08–146.00	0.5210
Polypharmacy (≥5 types of prescribed medicines)	1.20	0.28–5.27	0.8050
Nutritional intake status: Food diversity (good)	0.83	0.21–3.29	0.7910
Hearing impairment (1 point increase)	4.09	1.85–9.03	0.0005
Frailty status			
Robust	1.00		
Pre-frailty	1.62	0.10–25.70	0.7340
Frailty	20.40	1.48–280.00	0.0242

C statistic = 0.955 (0.931–0.980). Hosmer–Lemeshow test: chi-squared = 2.228, df = 8, *p* = 0.9732. OR: odds ratio, 95% C.I.: 95% confidence interval. All variables shown in the table were included in the analysis of the logistic regression model as independent or adjustment variables.

**Table 3 ijerph-20-04437-t003:** Associations of hearing impairment and frailty status with cognitive decline by the interaction term model.

Variables	OR	95% C.I.	*p* Value
Age (y)	1.04	0.92–1.17	0.5340
Body mass index (kg/m^2^)	1.25	0.99–1.58	0.0636
Sex (female)	0.54	0.11–2.57	0.4380
Financial status (poor)	0.70	0.15–3.18	0.6460
Living arrangement (alone)	0.26	0.03–1.95	0.1890
Social isolation (<12 points in 6LSNS)	1.67	0.32–8.70	0.5450
Number of social activities (0–6 types)	0.68	0.39–1.20	0.1860
Subjective health (poor)	0.60	0.11–3.20	0.5480
Medical history: Hypertension	1.02	0.22–4.68	0.9750
Medical history: Diabetes mellitus	1.84	0.36–9.39	0.4620
Medical history: Cerebrovascular disease	2.04	0.05–83.00	0.7070
Polypharmacy (≥5 types of prescribed medicines)	1.26	0.28–5.61	0.7630
Nutritional intake status: Food diversity (good)	0.82	0.21–3.22	0.7800
Interaction term			
Hearing impairment: Robust	2.57	0.83–7.96	0.1010
Hearing impairment: Pre-frailty	2.74	1.24–6.03	0.0125
Hearing impairment: Frailty	6.20	2.54–15.10	0.0001

C statistics = 0.955 (0.933–0.978). Hosmer–Lemeshow test: chi-squared = 2.213, df = 8, *p* = 0.9738. OR: odds ratio, 95% C.I.: 95% confidence interval. All variables shown in the table were included in the analysis of the logistic regression model as independent or adjustment variables.

## Data Availability

Data can be provided on request from the corresponding authors.

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
