# Peer review of "The Combination of Hearing Impairment and Frailty Is Associated with Cognitive Decline among Community-Dwelling Elderly in Japan"

_ijerph, 2023, doi:10.3390/ijerph20054437_

Round 1

Reviewer 1 Report

Searching for relationships between changes in old age and the onset of cognitive decline is central to current research.  The paper can be  potentially seen as a  contribution in this direction.  The authors however are not clear on the meaning they give to the term “risk factor”. Caution is necessary for searching for associations among the  factors  called into play.

First, and given the nature of the topics addressed, authors across the paper should refer to cognitive decline while sometimes they shift to dementia ( see as an example page 5 line 284): it is actually cognitive decline the topic they should refer to and the level at which results should  exhibit relevance.

More crucially there is the issue of what actually authors analyzed. They, in fact, throughout the paper  refer to hearing difficulty but  they should  instead refer to subjective hearing difficulty. This is what they analyzed in concrete and although correlations have been seen between the tool used and objective measures of hearing this is the level at which the theoretical analysis should be set. What clinicians know and research studies show is that often  behind the experience of hearing difficulties there is an attentional difficulty and literature shows evidence of the relevance of attentional problems in the onset of cognitive decline.   Starting from the introduction to the discussion, the authors should then reconsider the paper in light of this  mentioned. The description of the tool does not help  readers understand the actual content addressed on the “hearing subjective experience.”  Additionally, it would be helpful to have a coding of the answers describing the “hearing difficulty experience” in more detail and with more explicit levels of functioning as for the other dimensions addressed. 

Reviewer 2 Report

ijerph-2198100 Manuscript review 

Thank you for this opportunity to review this manuscript examining the longitudinal relationship between frailty, cognitive impairment and hearing loss.   

Title and abstract:  

In the title the authors should specify the association rather than simply stating that there is one. 

Hearing difficulty should be better defined from the outset. This should be clearly defined in the manuscript.  

Further, hearing impairment is a more scientific term and should be used in preference.  

What defined cognitive decline? This should be specified in the abstract. 

Was hearing independently associated with decline – if so (and I presume it was), this should be stated explicitly in the abstract.  

Main Manuscript: 

Introduction:

In the introduction, the authors should specify that at present most treatments target symptoms, though recent studies suggest that disease modifying therapies may offer some efficacy.  

I would avoid saying that "Thus, an interaction exists between hearing difficulty and frailty" - this is not clear from the evidence. An interaction "may exist". This study is being conducted to examine if frailty modifies/influences the relationship between hearing impairment and dementia/cognitive decline - this should be expressed more clearly at the end of the introduction, so the objective/a priori hypothesis is explicit for readers.

Methods:

Please define a housing estate

Did the inclusion criteria specify that patients had to be independent in all activities of daily living (ADL)? This would imply that patients could not have dementia as this by definition implies functional impairment in ADL. Is this correct? If not, this needs to be clarified.

As said, hearing difficulty should be defined aside from details on what was used (i.e. Questionnaire for Hearing) to assess it. 

Results:

The results are clearly presented.

To be clear, are the results in table 3 adjusted to account for the effects of age and other potential confounders that may mean that it is age that explains the relationship between these three strongly age-correlated conditions (i.e. hearing loss, cognitive decline and frailty)? From my read Table 2 presented the unadjusted ORs? This should be made clearer in the Table headings.

Does the effect show a clear incremental gradient i.e. for every one point increase in the KCL does the odds of cognitive decline increase and by what amount with and without the presence of hearing impairment? 

Discussion:

The discussion is well-written - please specify that given the poor response rate, there is likely selection bias which may have influenced the findings. 

Why do the authors think there was such a low response? This should be mentioned in the discussion. 

The population is likely to have been homogenous and this would reduce the representativeness of the findings i.e. externally generalisability such that there is a need to replicate these findings in bigger and more representative samples. 

Specify how this paper differs from ref 35: Bonfiglio, V.; Umegaki, H.; Kuzuya, M. A study on the relationship between cognitive performance, hearing impairment, and 426 frailty in older adults. Dement Geriatr Cogn Disord. 2020, 49, 156-162. https://doi.org/10.1159/000507214. I presume this relates to the difference between a survey of cognition and more objective testing. 
